# Assessment of pediatric surgical needs, health-seeking behaviors, and health systems in a rural district of Pakistan

**Saqib Hamid Qazi[1], Syed Saqlain Ali Meerza[2], Reinou S. Groen[3], Sohail Asghar Dogar[1], Mushtaq Mirani[2], Muhammad Khan Jamali[2], Zahid Ali Khan[2], Zahra Ali Padhani[2], Rasool Bux[4], Imran Ahmed Chahudary[4], Arjumand Rizvi[4], Saleem Islam[5], Sadaf Khan[6], Rizwan Haroon Ur Rashid[7], Syed Akbar Abbas[8], Abdul Sami Memon[9], Sadia Tabassum[10], Bakhtawar Dilawar[1], Jai K. Das[2,4]***

**1** Section of Pediatric Surgery, Department of Surgery, Aga Khan University, Karachi, Pakistan, **2** Institute for Global Health and Development, Aga Khan University, Karachi, Pakistan, **3** Department of Obstetrics and Gynecology, Alaska Native Medical Center, Anchorage, Alaska, **4** Division of Women and Child Health, Aga Khan University Hospital, Karachi, Pakistan, **5** Department of Pediatric Surgery, University of Florida, Gainesville, Florida, United States of America, **6** Department of Surgery, Aga Khan University, Karachi, Pakistan, **7** Section of Orthopedic Surgery, Department of Surgery, Aga Khan University, Karachi, Pakistan, **8** Section of Head and Neck Surgery, Department of Surgery Aga Khan University, Karachi, Pakistan, **9** Department of Ophthalmology and Visual Sciences, Aga Khan University, Karachi, Pakistan, **10** Section of Dermatology, Department of Medicine, Aga Khan University, Karachi, Pakistan

* jai.das@aku.edu

**Data Availability Statement:** All data generated or analysed during this study are included in this

## Abstract

Surgical conditions are responsible for up to 15% of total Disability-Adjusted Life Years (DALY) lost globally. Approximately 4.8 billion people have no access to surgical care and this studies aim is to assess the surgical disease burden in children under the age of five years. We used Surgeons OverSeas Assessment of Surgical Need (SOSAS) and Pediatric Personnel, Infrastructure, Procedures, Equipment, and Supplies (PediPIPES) survey tools in Tando Mohammad Khan (TMK). A set of photographs of lesions were also taken for review by experts. All the data was recorded electronically via an android application. The current surgical need was defined as the caregiver's reported surgical problems in their children and the unmet surgical need was defined as a surgical problem for which the respondent did not access care. Descriptive analysis was performed. Information of 6,371 children was collected. The study identified 1,794 children with 3,072 surgical lesions. Categorization of the lesions by the six body regions suggested that head and neck accounted for the greatest number of lesions (55.2%) and the most significant unmet surgical need (16.6%). The chest region had the least unmet surgical need of 5.9%. A large percentage of the lesions were managed at a health care facility, but the treatment essentially consisted of mainly medical management (87%), and surgical treatment was provided for only 11% of lesions. The health facility assessment suggested that trained personnel including surgeons, anesthetic, or trained nurses were only available at one hospital. Basic procedures such as suturing and wound debridement were only performed frequently. This study suggests a high rate of unmet surgical need and a paucity of trained health staff and resources in this rural

published article and its supplementary information file.

**Funding:** The study was funded by Bill and Melinda Gates Foundation under grant number OPP 1148892 to JKD. The funders had no role in study design, data collection and analysis, decision to publish, or preparation of the manuscript. The authors declare that the research was conducted in the absence of any commercial or financial relationships that could be construed as a potential conflict of interest.

**Competing interests:** The authors have declared that no competing interests exist.

setting of Pakistan. The government needs to make policies and ensure funding so that proper trained staff and supplies can be ensured at district level.

## Introduction

Surgical conditions are responsible for up to 15% of total disability-adjusted life years (DALYs) lost globally [1]. Worldwide estimates have found that approximately 4.8 billion people have no access to surgical care, and within South Asia, greater than 95% of the population do not have access to care for conditions that require surgical management [2]. Considering that greater than 50% of the population in the least developed regions worldwide is children, we can surmise that the surgical burden amongst children in Low and Low Middle-Income Countries (LMICs) is immense [3, 4]. Currently, a large disproportion exists between the wealthiest and poorest third of the population globally, with the wealthy receiving a major share of 73.6% of surgical procedures and the poor receiving only 3.5% [5]. Within poorer countries, surgical services are concentrated almost wholly in cities and reserved largely for those who can pay for them out of pocket [1].

Until recently, pediatric surgical care in low and middle-income countries (LMICs) was largely overlooked, with global health attention primarily addressing communicable diseases, and maternal and infant mortality [5]. However, improvement of surgical care delivery for children is now being prioritized as a fundamental component of health care in LMICs [6]. Improving surgical care delivery also has significant economic and welfare benefits for the population, as untreated surgical conditions increase medical costs, disability, and death [4]. Hence, the development of methods to enhance the quality of pediatric surgical care in low-resource regions can remarkably decrease childhood morbidity and mortality [7] and alleviate the associated financial and emotional stress.

To promote the development of LMICs healthcare delivery systems that includes pediatric surgical care, it is vital to understand the local burden of surgical disease in children, the local population's understanding of these disease pathologies, and the capacity of healthcare facilities to handle the burden of disease. It is crucial to understand the pediatric surgical health-related behaviors of the community, as children often cannot express accurately their health issues, and their health-related decisions are mostly handled by parents and caregivers. Therefore community-level surveys provide a thorough measure of real-time surgical need [4]. The Surgeons OverSeas Assessment of Surgical Need (SOSAS) survey has been the most frequently used population-based survey to estimate the surgical disease burden. It was primarily developed to measure surgical disease burden in LMICs [8]. From the time of its creation and first use, SOSAS has become a validated tool to assess a vast range of surgically treatable conditions at the population level [8–10]. It has previously been used in countries like Uganda, Rwanda, Nepal, and Sierra Leone [8–15]. The Pediatric Personnel, Infrastructure, Procedure, Equipment, and Supplies (PediPIPES) survey tool is designed to assess the pediatric surgical capacity of health facilities. This survey originated as the World Health Organization (WHO) tool for situational analysis but was subsequently redesigned by Surgeons OverSeas (SOS) to include absolute numbers of hospital beds and operating rooms, a binary system of measurement to allow easier counting of items, removing reasons for not performing procedures, and restructuring of individual questions [16].

Pakistan, a South Asian country with a population of approximately 221 million people is an LMIC with a Gross National Income (GNI) per capita of $1270 [17]. Approximately 63% of

the population lives in rural areas [18]. Several studies have been published describing childhood surgical emergencies in Pakistan. However, through a literature review, it was identified that no community-based survey tool has yet been employed to determine Pakistan's pediatric surgical disease burden in rural areas [19–21].

The primary aim of this study is to conduct a population-based study to assess the burden of surgical disease for children under the age of five years, the population's health-seeking behavior, and assess the health facilities in the rural district of Tando Mohammad Khan (TMK) in Sindh, Pakistan. It will contribute to our comprehension of the epidemiology of surgical diseases in rural Sindh and to an extent in Pakistan overall. It can help bring policymakers attention to the importance of improving access to surgical care facilities.

## Methodology

The SOSAS and PediPIPES were conducted in a representative population of children under the age of five years over a period of three months, between 22nd November 2019 and 28th February 2020.

### Study setting

Sindh is the second largest province by population (approximately six million people) and the third largest province in Pakistan [22]. TMK is one of the 29 districts in Sindh with an area of 1,814 square kilometers ($km^2$), an overall population of 677,228, and a population density of 373 people/$km^2$, Sindhi as the most commonly spoken language [23]. TMK district was chosen as the site for our project due to our established office in the district, the availability of skilled manpower and other resources to conduct this large-scale survey. TMK is also a representative of any rural district in Sindh.

### Survey tools

The SOSAS survey consists of two portions. The first section collects demographic details, including the age and sex of household members, and number of deaths in the household within the past year. Household members are defined as people living in the same physical structure. The second half of the survey gathers information from caregivers on both current and previous surgical conditions of their children, which is categorized into six distinct anatomical regions: face, head, and neck; chest and breast; abdomen; groin, genitals, and buttocks; back; arms, hands, legs, and feet. The caregivers answered questions based on their recall and their perception of a surgical condition. Additional questions cover the type of injury/accident, timing of the condition, and health-seeking behavior, which includes the type of health care sought, type of health care received, and reasons if care was not accessed. The survey questions were translated into Sindhi, the primary language of TMK by coordination of experts in both English and Sindhi language and was pilot tested in the field.

The PediPIPES survey assesses gaps in the availability of essential and emergency surgical care (EESC) at the district health facilities. The data items were divided into five sections: Personnel; Infrastructure; Procedures; Equipment; and Supplies. PediPIPES scores were calculated by allocating 1 point for each personnel, infrastructure, procedures, equipment, and supplies present in the facility and summing it. This number was then divided by the total number of data items (118) and multiplied by 10 to create the PediPIPES index. There is no maximum number for the PediPIPES index. This index indicates the capacity of health facilities to provide EESC to infants and children.

### Data collection

Households in TMK districts were line-listed and a total of 3,643 eligible houses were randomized. The SOSAS survey was conducted by research personnel who were trained for data collection and were monitored by senior managers. A pilot study for trial and improvements was conducted in 50 households. Verbal consent was obtained from the caregivers, and surveys were administered in Sindhi language. Caregivers provided the survey information about their children below the age of five years. The information was recorded electronically via an application developed specially for this survey by the District Monitoring Unity (DMU). The application was developed for Android, and an IIS-10 webserver was setup with MySQL database to collect data in the remote areas. APIs were developed on webserver in PHP to send and receive data to and from devices. The purpose of the android-based survey application was to assist data collectors in conducting interviews in the remote field areas and capture data in electronic format. The application was used on Samsung tab A7 tablet. To restrict inappropriate access to the application, authorized data collectors were required to authenticate using login credentials that were specifically generated for the use of this application on the server.

The PIPES survey data from all facilities in TMK was collected. The data was collected on hard copies and then entered onto an excel sheet. A total of 233 photographs were also taken of children with lesions to understand how these could help diagnosis and future roll-out of such a strategy. These were reviewed by the relevant surgeons at the Aga Khan University Hospital (AKUH) for diagnosis, and confirmation of the previously established diagnosis. These pictures were taken via the same application developed and stored on the tablet/phone under password protection.

### Data analysis

The current surgical need was defined as the caregivers reported surgical problems in their children, either present at the time of the surveys, or in the past. Unmet surgical need was defined as a current surgical problem for which the respondent did not access care for their child. Descriptive analysis was performed using STATA 16 (Stata Statistical Software: Release 16. College Station, TX: StataCorp LP).

### Ethics statement

The research was approved by the Ethical Review Committee (ERC) of the Aga Khan University (AKU) and the National Bioethics Committee (NBC). Confidentiality of all collected data was assigned high priority at each stage of data handling. The research participants were informed about the purpose, methods and benefits and intended uses of the research. Informed verbal consent was obtained. Respondents were free to stop interviews at any time or skip any questions they did not want to answer. They had the right to ask questions at any point before, during or after the interview. All interviews were conducted by trained personnel and in conditions of privacy. The respondents were informed about their rights. All data files were password-protected, and data was encrypted.

## Results

A total of 3,643 households were surveyed for children's surgical needs in TMK. Of these households, 3,821 care givers filled the Surgeons Overseas Assessment of Surgical Need (SOSAS) survey form. Survey identified a sum of 6,071 children, with majority in the age range 1–5 years (81.4%). A near equal male (50.5%) to female ratio (49.5%) was present.

## Burden

**Deaths.** According to our survey, 8% households reported death of a child in the last five years. Of these, 95.9% reported one death, 3.8% reported two deaths and 0.3% reported three deaths. Overall, 300 child deaths were recorded, with most occurring in the age range 1–12 months (79.3%) and a near equal occurrence in both genders [Male (56.9%) and Female (43.1%)]. Of the total, 36.3% deaths occurred due to a probable surgical condition, of which majority were reported to have occurred due to a congenital deformity (30.3%), or due to maternal bleeding or illness during the birth process (23.9%).

Most of these children with surgical conditions (79.8%) were taken to a health care facility [private facility (67.8%) or government facility (29.9%)] for treatment prior to death, while others received management at home (2.3%). The treatment received was focused primarily on medical management (96.6%). Only two patients (2.3%) underwent a major procedure and one (1.1%) underwent a minor procedure.

Most child deaths were reported at home (48.6%), followed by healthcare facility (42.2%). The most common reason identified for not taking the child to a health care facility was lack of time (child died before any arrangement could be made, 40.9%). Other reasons included the misconception of a condition being non-surgical (18.2%), or no money to avail care at health facility (13.6%) (**Table 1**).

**Lesions.** Of the 6,071 children included in the survey, a total of 1,794 (29.6%) children were identified to have or previously have had a total number of 3,072 lesions requiring surgical care, as some children were identified to have more than one type of lesion. Lesions were approximately equally distributed among males (52.6%) and females (47.4%). Most of the children affected were in the age range of 1–5 years (90.6%). An overview of the lesions is presented in **Table 2**. Through a head-to-toe inquiry via the SOSAS survey form, the head and neck region (Comprising of the head, eye, ear, and neck) was found to be most frequently affected (n = 1,697, 55.2%) followed by extremities (n = 524, 17.1%). By far, the most common lesions identified were non-injury-related wounds (37.6%) followed by mass/growth (13.6%). In majority of the cases (82.5%), lesions were not secondary to an accident. Amongst the remaining of 17.5% who suffered an accident; falls were the most common cause of injury (49.1%), followed by hot object/ hot liquid- related burn injury (23.4%).

**Head and neck.** A total of 1697 were lesions identified in this region. Of these, the most affected region were the ears (41.1%) with the highest percentage of lesions, followed by head (22.6%), face (17.5%), eyes (12%), and neck (6.8%). Lesions of the ear and face were mostly non-injury related, while the lesions on the head and the neck were primarily a mass or growth. Congenital deformity was the most common cause of the lesion on the eye. The unmet surgical need was found to be the highest in the head and neck region, contributing to 16.6% of the total lesions (**Table 3**).

**Chest.** In the 102 lesions identified on the chest, 23.5% comprised of a mass or growth, closely followed by non-injury related wounds (20.6%). Most lesions were not secondary to an accident (81.4%). Amongst the remaining 18.6% that occurred following an accident, falls were identified as one of the main reasons (52.6%). The unmet surgical need of lesions on the chest was lowest, 5.9%.

**Back.** We identified 87 lesions on the back, of which approximately half consisted of a mass or growth (50.6%). Only 13.8% lesions started after an injury/accident, while the remaining 86.2% lesions were not secondary to an accident. In the 12 injury related wounds; burns from hot liquid or object were the most common cause (41.7%). The unmet surgical need of lesions on the back was 6.9%.

**Table 1. Details about child deaths in the district.**

| Total households | | | **3,643** |
|---|---|---|---|
| Deaths in how many households in last 5 years | | | 290 (8%) |
| Number of deaths per household | One | | 278 (96%) |
| | Two | | 11 (3.8%) |
| Total number of deaths | | | 300 |
| Age at death | 0–1 month | | 238 (79.3%) |
| | 1–12 month | | 60 (20%) |
| | 1–5 years | | 2 (0.7%) |
| Gender of the child died | Male | | 159 (53%) |
| | Female | | 141 (47%) |
| Total number of problems 1 week prior to deaths | | | 300 |
| Surgical conditions | | | 109 (36.3%) |
| Types of surgical conditions | Injury | | 2 (0.7%) |
| | Wound not due to an injury | | 7 (2.3%) |
| | Bleeding or ill around childbirth | | 26 (8.7%) |
| | Mass (Growth or Swelling) | | 15 (5.0%) |
| | Deformity (Congenital) | | 33 (11.0%) |
| | Deformity (Acquired) | | 4 (1.3%) |
| | Abdominal distention or pain | | 22 (7.3%) |
| Non-surgical conditions | | | 191 (63.7%) |
| Total number of children with surgical lesions | | | 109 |
| Children taken to a health care facility | Yes | | 87 (79.8%) |
| | No | | 22 (20.2%) |
| | Reasons for not taking | No money for health care | 3 (13.6%) |
| | | No money for transportation | - |
| | | No time (Died before arrangements) | 9 (41%) |
| | | Fear / No trust | 2 (9.1%) |
| | | Not available (Facility/personnel/equipment) | 0 |
| | | No need (Condition is not surgical) | 4 (18.2%) |
| | | Other | 4 (18.2%) |
| Where were these conditions managed? | At home | | 2 (2.3%) |
| | Government health facility | | 26 (29.9%) |
| | Private health facility | | 59 (67.8%) |
| | NGO health facility | | 0 |
| | Hakeem/Traditional Healer | | 0 |
| Type of treatment received for surgical conditions | None/No surgical care | | 0 |
| | Only medical treatment | | 84 (96.6%) |
| | Major procedure (*A procedure that requires regional or general anesthesia*) | | 2 (2.3%) |
| | Minor procedure (*Dressings, wound care, punctures, suturing, and I&D*) | | 1 (1.1%) |
| | Manipulation / casting / sling | | 0 |
| | Traction | | 0 |
| Place of death | Home | | 53 (48.6%) |
| | Health Facility | | 46 (42.2%) |
| | On the way to the facility | | 9 (8.3%) |
| | Somewhere else | | 1 (0.9%) |

**Table 2. Details of children with lesions.**

| | | |
|---|---|---|
| Total number of children | | 6071 |
| Total number of children with surgical lesions | | 1794 (29.6%) |
| Total number of surgical lesions | | 3072 |
| Age | month | 5 (0.3%) |
| | 1–12 months | 192 (10.7%) |
| | 1–5 years | 1,597 (89.0%) |
| Gender of the child | Male | 939 (52.3%) |
| | Female | 855 (47.7%) |

**Abdomen.** In the 493 abdominal problems noted, majority consisted of a child being unable to urinate (47.9%) followed by abdominal distention or pain (34.1%). A total of 481 (97.6%) lesions were not preceded by an accident. Wounds secondary to an accident mostly resulted from burns with hot objects (41.7%). The unmet surgical need of lesion on abdomen was 13.4%.

**Buttocks/Groin/Genitalia.** In a total of 169 lesions identified, majority were not associated with injury (27.2%), followed by solid mass or growth (21.3%) (**Table 4**). Of the 19 lesions secondary to an accident or injury, hot object burns accounted for the most lesions (42.1%), followed by falls (21.1%), animal attacks (15.8%), and open fire explosions (15.8%). The unmet surgical need for the region was 14.8%.

**Extremities.** In the 296 lesions on the upper limb, proximally to distally; 13.9% were on the upper arm, 41.9% on the lower arm, 21.3% on the thumb and hands and 23% were on the fingers. Majority lesions were due to acquired deformities. Of the lesions secondary to injury/accident, falls were identified as the most common cause in the upper arm, lower arm, and hand/thumb, whereas for fingers hot object related burns accounted for the highest percentage of the total lesions.

Overall, 228 (9.3%) lesions were found on lower limbs. Proximally to distally; upper leg (20.2%), lower leg (22.8%), and feet (57%). Lesions of the upper legs were mainly deformities, while the lower legs were mostly affected by abnormal growths. Burns were the most frequently occurring lesions of the feet. The unmet surgical need for the combined upper and lower limb lesions was 11.8% (**Table 5**).

**Photographs.** A sum of 233 photographs were taken of children with lesions. These pictures were shared with surgeons at the Aga Khan University Hospital for review, and reconfirmation of preliminary diagnoses (diagnosis made in health facilities of TMK) via

**Table 3. Lesions on head and neck.**

| Region | Eyes | | Ears | | Face | | Head | | Neck | |
|---|---|---|---|---|---|---|---|---|---|---|
| Problem | Children | Lesions | Children | Lesions | Children | Lesions | Children | Lesions | Children | Lesions |
| Wound injury related | 16 (8.2%) | 18 (8.8%) | 19 (3.6%) | 20 (2.9%) | 61 (22.1%) | 61 (20.5%) | 57 (16.1%) | 60 (15.7%) | 7 (6.3%) | 7 (6.0%) |
| Wound not injury related | 64 (32.7%) | 69 (33.8%) | 453 (86.8%) | 616 (88.4%) | 98 (35.5%) | 111 (37.4%) | 123 (34.8%) | 135 (35.2%) | 42 (37.5%) | 42 (36.2%) |
| Burn | 2 (1.0%) | 2 (1.0%) | 0(0) | 0(0) | 14 (5.1%) | 14 (4.7%) | 2 (0.6%) | 2 (0.5%) | 2 (1.8%) | 2 (1.7%) |
| Mass or growth (solid) | 11 (5.6%) | 12 (5.9%) | 21 (4.0%) | 26 (3.7%) | 81 (29.3%) | 88 (29.6%) | 146 (41.4%) | 160 (41.8%) | 49 (43.8%) | 53 (45.7%) |
| Deformity congenital | 80 (40.8%) | 80 (39.2%) | 20 (3.8%) | 25 (3.6%) | 11 (4.0%) | 11 (3.7%) | 13 (3.7%) | 13 (3.4%) | 5 (4.5%) | 5 (4.3%) |
| Deformity acquired | 10 (5.1%) | 10 (4.9%) | 6 (1.1%) | 6 (0.9%) | 11 (4.0%) | 12 (4.0%) | 12 (3.4%) | 13 (3.4%) | 7 (6.3%) | 7 (6.0%) |
| Blindness | 1 (0.5%) | 1 (0.5%) | 0(0) | 0(0) | 0(0) | 0(0) | 0(0) | 0(0) | 0(0) | 0(0) |
| Reduced vision | 12 (6.1%) | 12 (5.9%) | 0(0) | 0(0) | 0(0) | 0(0) | 0(0) | 0(0) | 0(0) | 0(0) |
| Hearing loss | 0(0) | 0 (0) | 3 (0.6%) | 4 (0.6%) | 0(0) | 0(0) | 0(0) | 0(0) | 0(0) | 0(0) |

**Table 4. Lesions on chest, back, abdomen, Buttocks/Groin/Genitalia.**

| Region | Chest | | Back | | Abdomen | | Buttocks/Groin/Genitalia | |
|---|---|---|---|---|---|---|---|---|
| Problem | Children | Lesions | Children | Lesions | Children | Lesions | Children | Lesions |
| Wound injury related | 6 (6.0%) | 6 (5.9%) | 2 (2.6%) | 2 (2.3%) | 2 (0.5%) | 2 (0.4%) | 4 (2.5%) | 4 (2.4%) |
| Wound not injury related | 20 (20.0%) | 21 (20.6%) | 23 (29.5%) | 25 (28.7%) | 14 (3.2%) | 16 (3.2%) | 43 (26.7%) | 46 (27.2%) |
| Burn | 4 (4.0%) | 4 (3.9%) | 7 (9.0%) | 7 (8.0%) | 7 (1.6%) | 7 (1.4%) | 13 (8.1%) | 13 (7.7%) |
| Mass or growth (solid) | 23 (23.0%) | 24 (23.5%) | 37 (47.4%) | 44 (50.6%) | 10 (2.3%) | 10 (2.0%) | 32 (19.9%) | 36 (21.3%) |
| Mass or growth (reducible) | 0(0) | 0(0) | 0(0) | 0(0) | 16 (3.7%) | 16 (3.2%) | 14 (8.7%) | 14 (8.3%) |
| Deformity congenital | 13 (13.0%) | 13 (12.7%) | 5 (6.4%) | 5 (5.7%) | 14 (3.2%) | 15 (3.0%) | 16 (9.9%) | 16 (9.5%) |
| Deformity acquired | 8 (8.0%) | 8 (7.8%) | 4 (5.1%) | 4 (4.6%) | 4 (0.9%) | 4 (0.8%) | 4 (2.5%) | 4 (2.4%) |
| Foreign body | 13 (13.0%) | 13 (12.7%) | 0(0) | 0(0) | 0(0) | 0(0) | 0(0) | 0(0) |
| Cardiac (congenital) | 13 (13.0%) | 13 (12.7%) | 0(0) | 0(0) | 3 (0.7%) | 3 (0.6%) | 0(0) | 0(0) |
| Abdominal distention or pain | 0(0) | 0(0) | 0(0) | 0(0) | 105 (24.0%) | 116 (23.5%) | 0(0) | 0(0) |
| Inability to urinate | 0(0) | 0(0) | 0(0) | 0(0) | 202 (46.2%) | 236 (47.9%) | 0(0) | 0(0) |
| Renal stone | 0(0) | 0(0) | 0(0) | 0(0) | 20 (4.6%) | 20 (4.1%) | 0(0) | 0(0) |
| Bleeding (per rectum) | 0(0) | 0(0) | 0(0) | 0(0) | 36 (8.2%) | 41 (8.3%) | 0(0) | 0(0) |
| Bleeding (per penis | 0(0) | 0 (0) | 0(0) | 0(0) | 4 (0.9%) | 7 (1.4%) | 16 (9.9%) | 17 (10.1%) |

interdepartmental collaboration of multiple surgery faculty members. Majority of the pictures were of children with lesions on the head and neck region, followed by extremities and least from children with lesions on the back. More information about the diagnosis can be found in S1 Table.

## Health seeking behavior

A large percentage of the lesions (n = 2,632, 85.7%) were managed at a health care facility, ranging from private (52.4%), government (33.8%), and NGO-based setup (0.2%), while the remaining 7.8% lesions were managed at home, or by a Hakeem/traditional healer (5.9%). Treatment essentially consisted of mainly medical management (n = 2,289, 87%). Surgical treatment was only received for 291 (11%) lesions, which included minor procedures, major

**Table 5. Lesions on extremities.**

| Region | Fingers | | Thumb/ Hand | | Upper arm | | Lower arm | | Upper leg | | Lower leg | | Foot | |
|---|---|---|---|---|---|---|---|---|---|---|---|---|---|---|
| Problem | Children | Lesions | Children | Lesions | Children | Lesions | Children | Lesions | Children | Lesions | Children | Lesions | Children | Lesions |
| Wound injury related | 9 (13.8%) | 10 (14.7%) | 5 (8.3%) | 5 (7.9%) | 9 (25.7%) | 10 (24.4%) | 22 (20.8%) | 23 (18.5%) | 5 (11.1%) | 5 (10.9%) | 4 (8.5%) | 4 (7.7%) | 15 (12.1%) | 16 (12.3%) |
| Wound not injury related | 5 (7.7%) | 5 (7.4%) | 8 (13.3%) | 9 (14.3%) | 7 (20.0%) | 11 (26.8%) | 9 (8.5%) | 14 (11.3%) | 7 (15.6%) | 7 (15.2%) | 12 (25.5%) | 14 (26.9%) | 13 (10.5%) | 13 (10.0%) |
| Burn | 31 (47.7%) | 32 (47.1%) | 17 (28.3%) | 17 (27.0%) | 3 (8.6%) | 3 (7.3%) | 20 (18.9%) | 20 (16.1%) | 7 (15.6%) | 8 (17.4%) | 4 (8.5%) | 4 (7.7%) | 36 (29.0%) | 37 (28.5%) |
| Mass or growth (solid) | 4 (6.2%) | 4 (5.9%) | 5 (8.3%) | 5 (7.9%) | 3 (8.6%) | 3 (7.3%) | 11 (10.4%) | 12 (9.7%) | 6 (13.3%) | 6 (13.0%) | 13 (27.7%) | 15 (28.8%) | 15 (12.1%) | 18 (13.8%) |
| Deformity congenital | 4 (6.2%) | 4 (5.9%) | 5 (8.3%) | 5 (7.9%) | 0(0) | 0(0) | 4 (3.8%) | 4 (3.2%) | 11 (24.4%) | 11 (23.9%) | 5 (10.6%) | 5 (9.6%) | 24 (19.4%) | 25 (19.2%) |
| Deformity acquired | 12 (18.5%) | 13 (19.1%) | 20 (33.3%) | 22 (34.9%) | 13 (37.1%) | 14 (34.1%) | 39 (36.8%) | 50 (40.3%) | 9 (20.0%) | 9 (19.6%) | 9 (19.1%) | 10 (19.2%) | 20 (16.1%) | 20 (15.4%) |
| (Recurrent) drainage / discharge | 0(0) | 0(0) | 0(0) | 0(0) | 0(0) | 0(0) | 1 (0.9%) | 1 (0.8%) | 0(0) | 0(0) | 0(0) | 0(0) | 1 (0.8%) | 1 (0.8%) |

**Table 6. Lesion management and impact.**

| | | | |
|---|---|---|---|
| Total number of children with surgical conditions | | | 1794 |
| Total number of surgical conditions | | | 3072 |
| Lesion managed at health care facility | Yes | | 2632 (85.7%) |
| | No | | 440 (14.3%) |
| | Reasons for not visiting health facility | No money for health care | 135 (30.7%) |
| | | No money for transportation | 22 (5.0%) |
| | | No time (Died before arrangements) | 0 (0) |
| | | Fear / No trust | 36 (8.2%) |
| | | Not available (Facility/Personnel/Equipment) | 9 (2.0) |
| | | No need (Condition is not surgical) | 215 (48.9%) |
| | | Other | 23 (5.2%) |
| Where were these conditions managed? | At home | | 204 (7.8%) |
| | Government health facility | | 889 (33.8%) |
| | Private health facility | | 1,379 (52.4%) |
| | NGO health facility | | 4 (0.2%) |
| | Hakeem/Traditional Healer | | 156 (5.9%) |
| Type of treatment received for surgical conditions | None/No surgical care | | 52 (2.0%) |
| | Only medical treatment | | 2,289 (87.0%) |
| | Major procedure (*A procedure which requires regional or general anesthesia*) | | 22 (0.8%) |
| | Minor procedure (*Dressings, wound care, punctures, suturing and I&D*) | | 121 (4.6%) |
| | Manipulation / casting / sling | | 70 (2.7%) |
| | Traction | | 78 (3.0%) |
| Does this problem (Disability) still impact on child's daily life | The condition is not disabling | | 2,900 (94.4%) |
| | He/she feels ashamed | | 87 (2.8%) |
| | He/she not able to work like he/she used to | | 67 (2.2%) |
| | He/she needs help with transportation | | 116 (3.8%) |
| | He/she needs help with daily living | | 103 (3.4%) |

procedures, traction and manipulation/casting /sling. Fifty-two lesions did not receive any care (**Table 6**).

The main reasons for not seeking care included the perception of the condition being non-surgical (48.9%) and an inability to afford health care (30.7%).

## Health facility assessment

A total of 39 health facilities were identified and surveyed in the TMK district.

**Personnel.** Medical facilities were surveyed to identify presence of trained personnel to handle pediatric surgical care which included, pediatric and general surgeon, anesthesiologists, medical doctors, pediatricians, pediatric trained nurses and nurse anesthetics. Only one facility (district hospital) out of the 39 surveyed was identified to have trained personnel present, which comprised of three anesthesiologists, two pediatricians, two pediatric trained nurses and one general surgeon. There were no nurse anesthetists, pediatric surgeon or medical doctors available to operate on children in any other health facility.

**Infrastructure.** The 39 facilities comprised of 197 beds of which 22 were allocated for pediatric patients. All 39 facilities had an electricity supply (external source or generator). Other infrastructure components that were present in most of the facilities included running water and medical records. Laboratories to test blood and urine were available in 89.7% of the facilities surveyed (**Table 7**).

**Table 7. Infrastructure items.**

| Infrastructure | No. (%) * |
|---|---|
| Electricity (External source or generator) | 39 (100) |
| Running water | 38 (97.4) |
| Medical records | 38 (97.4) |
| Laboratory (Blood and urine) | 35 (89.7) |
| Incinerator | 9 (23.1) |
| Ultrasonography | 8 (20.5) |
| Plain radiography | 5 (12.8) |
| Postoperative care area | 5 (12.8) |
| Emergency department | 4 (10.3) |
| Special care baby unit | 4 (10.3) |
| Operating room | 3 (7.7) |
| Blood bank | 1 (2.6) |
| Pediatric ventilator | 1 (2.6) |

**Procedures.**   At the time of the survey, 14 different types of procedures were being performed at the health facilities. Basic and less intricate procedures were performed more than the complex procedures. Suturing (94.9%), wound debridement (94.9%) and incision and drainage of abscess (82.1%) were frequently performed (**Table 8**).

**Equipment and supplies.**   More than half of the hospitals had basic equipment and supplies as detailed in **Tables 9** and **10.** However, of note, apnea monitors, laparoscopic surgery supplies, chest tubes, neonatal T-piece and endoscopes were not available at any of the facilities. The availability of anesthesia machines, electrocautery machine, scalpel blades, syringe pumps, and sterile drapes was also significantly low (under 10% of facilities) (**Tables 9** and **10**).

**PediPIPES scores and indices.**   The hospital with the highest PediPIPES index of 5.7 in TMK had the following: Personnel score of 8, infrastructure score 9, procedure score 12, equipment score 17 and supplies score of 21. The hospital with the lowest index of 2.4 had the following; Personnel score of zero, infrastructure score 3, procedure score 3, equipment score 9 and supply score of 13. The mean PediPIPES index was found to be 3.0 in the 39 facilities surveyed.

## Discussion

This assessment of surgical burden in children under five years, unmet surgical needs, healthcare seeking behavior, and the health facility assessment in a rural district of Pakistan, is the first of its nature where a systematic tool is used to assess the surgical burden.

**Table 8. Procedures.**

| Anesthesia No. (%) | Respiratory No. (%) | Gastrointestinal No. (%) | Urogenital No. (%) | Orthopedic No. (%) | Unclassified No. (%) |
|---|---|---|---|---|---|
| Ketamine 1 (2.6) | Chest tube insertion 1 (2.5) | Appendectomy 1 (2.6) | Pediatric hernia repair 1 (2.6) | Traction of closed fracture 1 (2.6) | Resuscitation 8 (20.5) |
| General 1 (2.6) | | | Ovarian cystectomy 1 (2.6) | Casting of fracture 1 (2.6) | Suturing 37 (94.9) |
| Spinal 1 (2.6) | | | | | Wound debridement 37 (94.9) |
| | | | | | I&D of abscess 32 (82.1) |
| | | | | | Burn management 4 (10.3) |

**Table 9. Equipment.**

| Equipment | No. (%) |
|---|---|
| Thermometer | 38 (97.4) |
| Stethoscope | 38 (97.4) |
| Weighing scale (Infant) | 38 (97.4) |
| Oxygen mask and tubing | 38 (97.4) |
| Compressed oxygen in cylinder | 38 (97.4) |
| Pulse oximeter | 37 (94.9) |
| Kidney dish | 35 (89.7) |
| Sterilizer (Autoclave) | 32 (82.1) |
| Oxygen concentrator | 31 (79.4) |
| Suction pump (Manual or electric) | 24 (61.5) |
| Blood pressure measuring equipment (Pediatric cuff) | 20 (51.3) |
| Bag-valve mask (Pediatric) | 19 (48.7) |
| Oropharyngeal airway (Pediatric) | 7 (17.9) |
| Surgical instrument set (Abdominal) | 5 (12.8) |
| Endotracheal tube (Pediatric) | 5 (12.8) |
| Operating room light | 4 (10.3) |
| Anesthesia machine | 2 (5.1) |
| Syringe pumps | 2 (5.1) |
| Electrocautery machine | 1 (2.6) |

**Table 10. Supplies.**

| Supplies | No. (%) |
|---|---|
| Gloves (Examination) | 39 (100) |
| Adhesive Tape | 39 (100) |
| Apron | 39 (100) |
| IV cannulas | 38 (97.4) |
| Syringes | 38 (97.4) |
| Tourniquet | 38 (97.4) |
| Sterile gauze | 37 (94.9) |
| Bandages sterile | 37 (94.9) |
| Face masks | 37 (94.9) |
| Gloves (Sterile) | 36 (92.3) |
| Gowns (For surgeon/Scrub nurse) | 36 (92.3) |
| Suture (Non-absorbable) | 34 (87.2) |
| Boots (Theatre shoes) | 34 (87.2) |
| Disposable needles | 32 (82.1) |
| Suture (Absorbable) | 30 (76.9) |
| Intravenous fluid infusion sets | 24 (61.5) |
| Nasogastric tubes 12F or smaller | 20 (51.3) |
| Eye protection (Goggles, safety glasses) | 11 (28.2) |
| Blood transfusion sets | 10 (25.6) |
| Urinary catheters (Must include 6F) | 8 (20.5) |
| Sharps disposal container | 4 (10.3) |
| Drapes (For operations) | 1 (2.6) |

This study highlights the significant surgical burden and unmet need in a rural setting of Pakistan. The head and neck accounted for the greatest number of lesions (55.2%) and the most significant unmet surgical need (16.6%), followed by the buttocks/groin and genitalia and the chest region had the least unmet surgical need. A large number of lesions were managed at a health care facility, mostly private health facilities. The treatment mainly consisted of medical management and surgical treatment was only given for 11% of the lesions. Compared to other LMICs in South Asia such as India (unmet surgical need of 6.5%) and Nepal (unmet surgical need of 5%) [9, 10, 13, 15, 24], the rates of current and unmet surgical need in TMK are high (14.3%), and Pakistan currently faces a challenge of inadequate surgical care delivery to children [25]. According to the World Bank 2019 data, India a neighboring country, reported a lower number of physicians per 1000 individuals compared to Pakistan (0.7 vs. 1.1) [26]. However, despite this the SOSAS study conducted in 2019 by Cherukupalli et al. [5] reported an unmet surgical need of 6.5% in rural India, which is much lower than the 14.3% in TMK. Although this is a limited comparison since both studies were conducted in specific regions of India and Pakistan which may have been affected by multiple geographic and cultural factors, it suggests a need for larger surveys and subsequent intervention in Pakistan.

Amongst children who had died due to a surgical cause and children who currently face a surgical problem, a large proportion did visit a health facility but were not managed appropriately, highlighting gaps in the quality of health service delivery in the health facilities. The parallel PediPIPES survey helped us explore the health facilities in the region which identified that not only did the facilities have a shortage of equipment, they also lacked the presence of trained surgical and especially pediatric surgery staff. Appropriate government policies at both provincial and federal level are required for the hiring and training of pediatric surgical staff and budget allocation to upgrade the standard of the health facilities. Along with the upgradation of health facilities, community outreach, there is need for awareness campaigns on emergent and disability inducing surgical diseases so that prompt care is sought. This need is highlighted by the fact that a large proportion of patients who were not taken to a health facility were due to lack of awareness of the acute nature of the condition or the inability to recognize the need for surgical management.

Access to surgical care could be addressed with the continuing development of surgical capacity at lower-level facilities, improving referral systems, and surgical training of non-physicians. Countries such as Malawi, and Mozambique have invested in training of non-physicians in surgery, however only a small fraction of the need can be dealt with in this fashion [27]. In addition to health facility capacity development, community-based educational programs for strengthening pre-hospital systems are vital [5]. Although the trauma burden in children under five is low in TMK, an initiative to provide basic first responder courses [28] together with early identification and management of surgical cases could help lay-people to contribute to better care and can act as a step to improve pre-hospital infrastructure, and increasing awareness in community [28]. Health service delivery may be greatly improved through a simple reorganization of services without any cost. For example, this can be demonstrated by the essential trauma-care guidelines and their use as needs assessment tools in multiple countries [29]. Increasing the availability of medications, equipment, supplies, and banked blood at the primary and secondary health care centers are essential steps as well [4, 30]. This allows adequate care for basic surgical problems at lower-level hospitals, whilst also providing an opportunity to stabilize more critically ill patients prior to being transported to a tertiary care center for more complex procedures.

The study has some potential limitations that included the study site may not represent the overall national picture of surgical burden but offers significant representation of any rural setting of Sindh. One significant limitation is in-built in the survey's definition of operative

conditions, as all operative conditions in this survey relied upon participants' reported conditions in the verbal interview. Thus, the numbers identified through this survey serve as proxies in estimating operative disease prevalence. This may overestimate some conditions, which may require follow up rather than surgical intervention. The survey also does not determine diseases such as cancers, which could require surgical care, thus underestimating the true surgical burden of disease. Despite these limitations, surgical conditions recognized by SOSAS provide an initial estimate of the need for surgical consultations.

After this initial assessment in a rural district of Pakistan, the next step should be to perform a larger countrywide survey in Pakistan with an aim to compare the burden of surgical conditions in urban and rural regions. We also need to explore the differences in community's approach to health seeking and the treatment options available. The successful trial implementation of photographs taken on-site and using them to reevaluate the established diagnosis with the help of physicians at a tertiary care setup, from different surgical sub-specialties. Follow ups can be established to link the patients with surgical conditions to trained nurses and surgeons in urban centers for surgical assistance. While we know it will take time in upgradation the current rural health care setup, this may help the more acutely suffering patients. A larger more generalizable survey for assessment of health facility standards and the surgical burden is needed in the other provinces, for a better understanding and comparable results which will inform and assist policy makers for drafting a targeted and priority based solution to the problem that exists.

## Conclusions

The surveys in this study have only been conducted in one rural district and may not represent Pakistan's overall burden of pediatric surgical disease, but it does highlight a significant and high rate of unmet surgical need and gaps in pediatric surgery health services which needs to be addressed. This needs to be done both at the community level through increased awareness of surgical conditions, and at the government level by introducing policies and budget allocation for upgrading of health facilities.

## Supporting information

**S1 Table. Diagnoses of photographs taken at the study site.**
(DOCX)

## Author Contributions

**Conceptualization:** Saqib Hamid Qazi, Jai K. Das.

**Data curation:** Saqib Hamid Qazi, Rasool Bux.

**Formal analysis:** Syed Saqlain Ali Meerza, Rasool Bux.

**Funding acquisition:** Saqib Hamid Qazi, Zahra Ali Padhani, Jai K. Das.

**Methodology:** Syed Saqlain Ali Meerza.

**Project administration:** Saqib Hamid Qazi, Mushtaq Mirani, Muhammad Khan Jamali, Zahid Ali Khan.

**Resources:** Syed Akbar Abbas.

**Software:** Rasool Bux.

**Supervision:** Saqib Hamid Qazi, Mushtaq Mirani, Muhammad Khan Jamali, Zahid Ali Khan, Imran Ahmed Chahudary, Arjumand Rizvi, Jai K. Das.

**Validation:** Syed Saqlain Ali Meerza, Rasool Bux.

**Writing – original draft:** Syed Saqlain Ali Meerza.

**Writing – review & editing:** Saqib Hamid Qazi, Syed Saqlain Ali Meerza, Reinou S. Groen, Sohail Asghar Dogar, Saleem Islam, Sadaf Khan, Rizwan Haroon Ur Rashid, Abdul Sami Memon, Sadia Tabassum, Bakhtawar Dilawar, Jai K. Das.

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
