## [Decision Letter · Decision Letter 0]

9 Aug 2022

PGPH-D-22-01041

Assessment of pediatric surgical needs, health seeking behaviors and health systems in the rural district of Tando Mohammad Khan Sindh, Pakistan

Dear Dr. Das,

Thank you for submitting your manuscript to PLOS Global Public Health. After careful consideration, we feel that it has merit but does not fully meet PLOS Global Public Health’s publication criteria as it currently stands. Therefore, we invite you to submit a revised version of the manuscript that addresses the points raised during the review process.

We look forward to receiving your revised manuscript.

Kind regards,

Jagnoor Jagnoor

Academic Editor

Journal Requirements:

1. Please amend your detailed online Financial Disclosure statement. This is published with the article. It must therefore be completed in full sentences and contain the exact wording you wish to be published.

Please state the initials, alongside each funding source, of each author to receive each grant.

2. Please update your online Competing Interests statement. If you have no competing interests to declare, please state: “The authors have declared that no competing interests exist.”

Additional Editor Comments (if provided):

Thank you for your submission. We have now received three reviewer comments as below. We hope these insights are helpful in revising the manuscript.

Reviewers' comments:

Reviewer's Responses to Questions

**Comments to the Author**

1. Does this manuscript meet PLOS Global Public Health’s publication criteria? Is the manuscript technically sound, and do the data support the conclusions? The manuscript must describe methodologically and ethically rigorous research with conclusions that are appropriately drawn based on the data presented.

Reviewer #1: Partly

Reviewer #2: Yes

Reviewer #3: Partly

2. Has the statistical analysis been performed appropriately and rigorously?

Reviewer #1: Yes

Reviewer #2: N/A

Reviewer #3: N/A

3. Have the authors made all data underlying the findings in their manuscript fully available (please refer to the Data Availability Statement at the start of the manuscript PDF file)?

Reviewer #1: Yes

Reviewer #2: Yes

Reviewer #3: Yes

4. Is the manuscript presented in an intelligible fashion and written in standard English?

Reviewer #1: No

Reviewer #2: Yes

Reviewer #3: Yes

5. Review Comments to the Author

Reviewer #1: Overall, this is an assessment of pediatric surgical needs, health seeking behaviors, and health systems in a rural district in Pakistan.

1. Publication criteria

a. Partly

2. Statistical analysis appropriate and rigorous?

a. Yes – descriptive statistics only in this study

3. Data fully available?

a. Yes – authors state the data is reported in the manuscript or in supplemental information

4. Intelligible and standard English?

a. No – there are some areas that require some editing including capitalization, punctuation, and grammar. I tried to note them below.

5. Review comments to author:

a. Title: I would consider saying “a rural district in Pakistan” rather than spelling out the district name. This would cut a lot of characters and simplify it a bit. Readers can see the specifics when they read the article, and the specific place is not needed in the title.

b. Abstract:

c. DALY please spell out on first mention with the acronym in parentheses.

d. On first mention of all acronyms, please spell out in full. Also spell out district name in full on first mention.

e. You mention the names of the survey tools but do not define them or describe what they are assessing. In the methods you need to describe these tools better, as not knowing in a general sense what they are measuring inhibits my ability to interpret the results section.

f. No need to include ethical approval in the abstract. This can be in the body of the manuscript instead.

g. Please spell out healthcare facilities in the district as I do not know what all the acronyms (like RHCs, BHUs, DCDs, etc) are.

h. In the methods you need at least a sentence about the statistical analysis, even if it was just saying that descriptive statistics were performed.

i. I would recommend better organization of the results section. Rather than listing the different areas of the body with frequencies, I would focus on the areas that you feel warrant mention the most, such as greatest number of lesions and most surgical need, as well as least. I would not list them all in a laundry list.

j. I would like a bit more explanation on what “unmet surgical need” means. How are you measuring that?

k. The result section of the abstract that discusses the health facilities is currently pretty meaningless without the definitions of what the different facilities are.

l. What does the PediPIPES score mean? Just a brief description would go a long way here.

m. The first sentence of the conclusion is an overstatement. I think I would say it holds great significance for one area of Pakistan, or even just cut the first sentence and go straight into the second sentence, “This study provides important insight into the birden of children’s operative disease in Pakistan’s rural district of TMK.”

n. Introduction:

o. Please write out LMICs at first mention and put acronym in parentheses (line 70).

p. In line 74 “reserved largely for those who can pay for them”, do you mean pay out of pocket or with insurance? Clarify here.

q. In line 84-85, you mention individual understanding of these disease pathologies. Do you mean patient understanding or healthcare provider understanding? Which of these are you measuring in this study?

r. Line 87 I would say “as children OFTEN lack the ability to express their health issues”, as many children are able to express their health issues quite well.

s. The paragraph from lines 83-97 would benefit from a little bit of organization, as it goes from Health-related decisions handled by parents and families to surveys rather abruptly. Consider a transition here.

t. In line 104-105, the first sentence of the last paragraph is too long and disjointed. I would cut “keeping in mind the global trends of surgical care delivery depicted in past research” and keep the rest the same.

u. Line 109 “form” should be plural.

v. How will your study lay the foundation for improving access to surgical care facilities? This seems like a bit of a reach to me.

w. Methodology

x. Line 114 survey should be plural since there were multiple surveys completed.

y. In the survey tool section, what if a mother was not present in a household to answer the survey? Were fathers or other caregivers allowed to answer?

z. In lines 145-146, I’m confused as to why you mention the 256 items of the WHO tool without describing the tool. Is it similar to the PediPIPES tool? Why did you choose PediPIPES and not the WHO tool?

aa. Lines 151-152, same comment about the mothers being the only respondents to the survey? Also, if mothers provided info about children under 5, who provided info for children above the age of 5? I see below that the study is in children under 5. This needs to be made clear much earlier in the paper.

bb. How was the info recorded electronically? On Excel? REDCap? More details here.

cc. I find it confusing that you give the numbers of health facilities in Sindh, but then only report on the 39 health facilities in the district. I don’t think you need the numbers of health facilities in Sindh, but rather just the district to keep things clear and simple.

dd. What is the difference between a research coordinator and research specialist? If not applicable to this study, you can probably just say two research personnel.

ee. Line 164 – the mention of the photographs seems random and has not been mentioned elsewhere. What was this for? How does this relate to the study?

ff. Data analysis section, lines 168-169 – when you say current surgical need was self-reported, who is self-reporting this? What interview? With families or health facilities?

gg. Results

hh. In the first paragraph, you summarize the demographics, but do not mention all the demographics. I found myself wondering where the demographics table was. Is there a reason you do not have a demographics table? I would consider adding one.

ii. You do not need to say “a rural district in Pakistan” in line 174 as this has been well described above.

jj. You included quite a bit of information about deaths, including a lot of info about deaths that are not surgical or related to surgical care. This manuscript is incredibly long, and this could be one way you could shorten it, but removing a lot of the non-surgical related data about deaths.

kk. In the description of the lesions, you report a lot of the same data both in text and in table form. I recommend cutting out a lot of repetition with having only the highly pertinent data (highest percentages, etc) in the text and then having all the data reported in the tables (like you do currently). As it is now, I am getting lost in the data and unable to make any meaningful conclusions. Hopefully by cutting down the long lists of percentages and frequencies in the text you will be able to only report the high yield results you want readers to gain from this section.

ll. Line 271 Photographs – Again, I’m unclear as to how the photographs relate to this study. I’m not sure I would include them as I don’t think they add to your study. I would remove them.

mm. Health seeking behavior section – line 282 please say treatment consisted of mainly medical management, as it was not only medical management.

nn. Line 289 – “children reported”, so children filled out the surgeries? Above you say parents filled them out for children under 5. Were these children over 5? This needs to be clarified as it’s unclear the age of the population and who is filling out each survey.

oo. Line 291 – children are reporting working? How old are these children?

pp. Table 4 – please review the capitalization and punctuation in this table and fix errors. An example is that “lesion” is not capitalized and “Health” is in the middle of a sentence. Please do this for all tables.

qq. Personnel – what personnel were present at the other hospitals, not just the DHQ?

rr. Infrastructure – this paragraph is centered and looks different than the others. Please fix the formatting here.

ss. Table 7 and 8 – it would be helpful to list the conditions with greatest frequency/percentage at the top and then with decreasing frequency. I think this would make the table easier to read. Also, the title should be “Equipment”.

tt. Equipment and Supplies – I find the listing of equipment to be a little repetitive and hard to read. You don’t need to list everything that is already present in the table. Just point out relevant results that you want readers to take away as important.

uu. PediPIPES scores – this section needs some major editing because I don’t know what a score of 8, 9, or 12 means. Listing numbers without telling us what they mean is useless. Also, which hospital had the highest scores that you are referring to? What scores did the other hospitals have? It’s hard to know what this section means without details and knowing what the scale of the scores are.

vv. Discussion

ww. The first two sentences of the discussion would belong better in the introduction as they give good info on the SOSAS survey and why you chose to use it.

xx. Lines 360-362 – this sentence about the primary aim of the study is repetitive with the primary aim in the intro. I would rephrase here.

yy. Lines 363-372 are a restatement of the results rather than summarizing the findings and discussing the findings. You are reporting your results section again here rather than comparing and contrasting with the existing literature.

zz. Again, lines 373-378 are a restatement of the results. This nicely distills the results and you could almost move this to the results section as I like it better than the long list of findings you currently have in the results, but this does not belong in the discussion. You need to discuss the findings, not restate them in the discussion section.

aaa. In line 382, you say compared to other LMICS, but don’t give the rates of surgical need in other LMICs. Report these rates, and compare and contrast them with your findings. I need more discussion of the existing literature in the discussion section. More like you currently have in lines 386-388!

bbb. The paragraph from lines 389-405 is excellent and has some great suggestions for improvement. I want more of this!

ccc. Were there any limitations to the health facility assessment part of this study? Who were performing the assessments of the health facilities? Is it possible there could have been errors in reporting capacity?

ddd. Lines 417-420 – the photographs are mentioned here again, but not elsewhere in the discussion. I still feel that removing the photographs is the right choice from this paper.

eee. I think that a stronger closing is needed. What are the next steps, and how does this study lead to them? I would close with that rather than a random comment about incorporating a physical examination.

Reviewer #2: Overall an important effort by the investigators. Few questions/ observations that came up during the review:

1. What was the reason behind selection of District TMK? How is this representative of the rural areas of Pakistan?

2. Are the results generalizable to other parts of the country/ region?

3. Instruments: the authors mention that these instruments are tested for the first time in Pakistan- were they tested for validity, inter-rater reliability?

4. This is a purely descriptive study with frequency table of presumed surgical conditions in pediatric population and availability of surgical care in the district. PIPES survey results demonstrate a general lack of access and availability of surgical services, referral system and questionable quality of available services. How these results could be used to a) quantify the magnitude and epidemiology of pediatric surgical conditions requiring surgical intervention or close follow up; b) informing decision makers about upgrading or expanding the services (specifically for pediatric surgery) ; c) support local communities in accessing and receiving appropriate surgical care and curative services; d) health system strengthening- also in terms of task shifting, referrals and infrastructure

5. It is better not to use abbreviations in the abstract for the first time.

Reviewer #3: Firstly, thank you for allowing me to review this wonderful work. The authors did quite well in assessing the burden of surgical disease in Pakistan. I appreciate the conceptualisation, claims and context of this work. Overall, I think this work would contribute to the body of knowledge if the following points as specified below were carefully taken into consideration.

Abstract

Page 4 ln 23/24 – needs some copyediting. Again, the justification for this study is not well presented here under abstract. The author can make it clearer by rephrasing.

Pg 4 ln 26/27; the author used least developed country and LMICs interchangeably. I advise the use of LMICs more so that Pakistan falls within this category according to world bank ranking.

The abstract did not state any findings on one of the main objectives of this study – health seeking behaviours.

Also, there is lack of consistency in using abbreviations and definition of abbreviations at first mention. For instance, whereas ERC, NBC, and AKU were defined earlier in abstract, others such as DALY, SOSAS and PediPIPES were only defined in the main body of the article.

Introduction

Except for copyediting in ln 65/66, the claims and objectives were well explained and contextualised. The conceptualisation of this idea is also novel. I agree with the authors that community assessment of surgical burden of disease in LMICs is critical to understanding the true picture of surgical needs to advocate for policy change or formulation.

Methodology

The authors can make the methodology more comprehensive. For example, there is not mention on when and how literature was searched, and on what data bases was the search conducted- albeit this is a primary study, it is still essential to let the readers know how you generated evidence from literature to justify this study. Again, if the literature search was done in 2019/2020 as at when the study was conducted, then, could new evidence have emerged?

I am also curious how the authors decided on conducting this research in TMK despite many other provinces in Sindhi of Pakistan. Was this systematically arrived at possibly through a scientific sampling technique? If yes, this can be explained further, please. Authors may wish to check this link for guidance on sampling technique

https://www.ncbi.nlm.nih.gov/pmc/articles/PMC5325924/

Further, I believe the authors can provide detailed information on the choice of only including under-5 children as study population despite the title and research justification suggesting paediatric population as a whole. In other words, a short paragraph to explain inclusion and exclusion criteria will be helpful. Again, any reasons why only mothers were interviewed instead of any head of household or guardian?

One may require that more information should be provided on the type of electronic method the data was collected onto – google form or Microsoft or excel etc

Kindly consider adding the common language of communication in TMK under ‘location’ subsection. Were these research associates trained in Sindhi given that the questionnaire was translated to this language?

Results

Tables were clearly labelled and presented. I appreciated the subcategorization of results on surgical needs. All tables were quite informative immediately.

Discussion

The authors could be more critical in exploring why despite going to the hospital, children were unattended to, accounting for high mortality. Is it lack of manpower or lack of supplies or infrastructure (ln 364-369)?

Ln 389-391 is assertive and would need reference(s). Kindly consider.

As part of limitation, the authors may also wish to consider if this sample size is adequate to generalise the evidence so generated from this study to Pakistan as a whole.

Conclusion:

This section is entirely missing in the main article. However, from the conclusion in the abstract, I am afraid, it appears quite generic and not entirely supported specifically by data of this study. I advise this should be addressed

6. PLOS authors have the option to publish the peer review history of their article (what does this mean?). If published, this will include your full peer review and any attached files.

**Do you want your identity to be public for this peer review?** For information about this choice, including consent withdrawal, please see our Privacy Policy.

Reviewer #1: No

Reviewer #2: No

Reviewer #3: No

---

## [Decision Letter · Decision Letter 1]

24 Nov 2022

PGPH-D-22-01041R1

Assessment of pediatric surgical needs, health-seeking behaviors, and health systems in a rural district of Pakistan.

Dear Dr. Das,

Thank you for submitting your manuscript to PLOS Global Public Health. After careful consideration, we feel that it has merit but does not fully meet PLOS Global Public Health’s publication criteria as it currently stands. Therefore, we invite you to submit a revised version of the manuscript that addresses the points raised during the review process.

We look forward to receiving your revised manuscript.

Kind regards,

Jagnoor Jagnoor

Academic Editor

Journal Requirements:

Additional Editor Comments (if provided):

Reviewers' comments:

Reviewer's Responses to Questions

**Comments to the Author**

1. If the authors have adequately addressed your comments raised in a previous round of review and you feel that this manuscript is now acceptable for publication, you may indicate that here to bypass the “Comments to the Author” section, enter your conflict of interest statement in the “Confidential to Editor” section, and submit your "Accept" recommendation.

Reviewer #2: (No Response)

Reviewer #3: All comments have been addressed

2. Does this manuscript meet PLOS Global Public Health’s publication criteria? Is the manuscript technically sound, and do the data support the conclusions? The manuscript must describe methodologically and ethically rigorous research with conclusions that are appropriately drawn based on the data presented.

Reviewer #2: Partly

Reviewer #3: Yes

3. Has the statistical analysis been performed appropriately and rigorously?

Reviewer #2: N/A

Reviewer #3: N/A

4. Have the authors made all data underlying the findings in their manuscript fully available (please refer to the Data Availability Statement at the start of the manuscript PDF file)?

Reviewer #2: Yes

Reviewer #3: Yes

5. Is the manuscript presented in an intelligible fashion and written in standard English?

Reviewer #2: Yes

Reviewer #3: Yes

6. Review Comments to the Author

Reviewer #2: I think authors have done a good job in addressing reviewer's comments and feedback. English language copyediting has made the manuscript more readable. The presentation of results and discussion has improved. The authors correctly recognized that children with a lesion and children with a lesion that requires surgical intervention are two different things, and this is a major limitation of the study methods. My residual concerns/ comments about this study are as follows:

This is a cross sectional study that could be labelled as a situational analysis.

The use of two different tools and population based survey methods are the strengths of study; however, nothing in the results is a news or a novel information for anyone doing health systems research in Pakistan or similar context.

Line 365- If the burden of trauma in under 5 is low then directing scarce is resources in training first responders is a waste. First responder courses may be needed in TMK, but not based on this data.

Line 369-372: I beg to differ from the authors- many countries have spent time, effort, money and capacity development for implementation of WHO trauma guidelines, only to find that random/ patchy implementation does not work until the entire pre-hospital and hospital based trauma system is targeted for improvement. Policy making and unrestricted funding are most important factors in brining a positive change in trauma care.

For the reasons mentioned above, conducting a nationwide surgical burden study (line 387) would only provide more numbers and tables. Further efforts should be directed in finding strategies to deliver care in austere environments, such as using Telehealth for surgical consultation and diagnosis, creating strong referral systems and effective capacity building of the existing staff.

Thanks and best of luck.

Reviewer #3: Dear Authors,

Thank you for considering our previous comment useful especially in the methodology aspect. I am excited that you are authoring a paper that will stimulate discussion among actors of this field. Well done!

7. PLOS authors have the option to publish the peer review history of their article (what does this mean?). If published, this will include your full peer review and any attached files.

**Do you want your identity to be public for this peer review?** For information about this choice, including consent withdrawal, please see our Privacy Policy.

Reviewer #2: No

Reviewer #3: No

---

## [Editor Report · Decision Letter 2]

7 Dec 2022

Assessment of pediatric surgical needs, health-seeking behaviors, and health systems in a rural district of Pakistan.

PGPH-D-22-01041R2

Dear Dr. Das,

We are pleased to inform you that your manuscript 'Assessment of pediatric surgical needs, health-seeking behaviors, and health systems in a rural district of Pakistan.' has been provisionally accepted for publication in PLOS Global Public Health.

Best regards,

Jagnoor Jagnoor

Academic Editor